

# Conformational heterogeneity of Savinase from NMR, HDX-MS and X-ray diffraction analysis

Shanshan Wu[1], Tam T.T.N. Nguyen[2], Olga V. Moroz[3], Johan P. Turkenburg[3], Jens E. Nielsen[4], Keith S. Wilson[3], Kasper D. Rand[2] and Kaare Teilum[1]

[1] Structural Biology and NMR Laboratory, Linderstrøm-Lang Centre for Protein Science, Department of Biology, University of Copenhagen, Copenhagen N, Denmark
[2] Protein Analysis Group, Department of Pharmacy, University of Copenhagen, Copenhagen Ø, Denmark
[3] York Structural Biology Laboratory, Department of Chemistry, University of York, York, United Kingdom
[4] Novozymes A/S, Lyngby, Denmark

Corresponding author
Kaare Teilum,
kaare.teilum@bio.ku.dk

## ABSTRACT

**Background**. Several examples have emerged of enzymes where slow conformational changes are of key importance for function and where low populated conformations in the resting enzyme resemble the conformations of intermediate states in the catalytic process. Previous work on the subtilisin protease, Savinase, from *Bacillus lentus* by NMR spectroscopy suggested that this enzyme undergoes slow conformational dynamics around the substrate binding site. However, the functional importance of such dynamics is unknown.

**Methods**. Here we have probed the conformational heterogeneity in Savinase by following the temperature dependent chemical shift changes. In addition, we have measured changes in the local stability of the enzyme when the inhibitor phenylmethylsulfonyl fluoride is bound using hydrogen-deuterium exchange mass spectrometry (HDX-MS). Finally, we have used X-ray crystallography to compare electron densities collected at cryogenic and ambient temperatures and searched for possible low populated alternative conformations in the crystals.

**Results**. The NMR temperature titration shows that Savinase is most flexible around the active site, but no distinct alternative states could be identified. The HDX shows that modification of Savinase with inhibitor has very little impact on the stability of hydrogen bonds and solvent accessibility of the backbone. The most pronounced structural heterogeneities detected in the diffraction data are limited to alternative side-chain rotamers and a short peptide segment that has an alternative main-chain conformation in the crystal at cryo conditions. Collectively, our data show that there is very little structural heterogeneity in the resting state of Savinase and hence that Savinase does not rely on conformational selection to drive the catalytic process.

## INTRODUCTION

Subtilisins are serine proteases with broad substrate specificity that have found widespread use in the laundry detergent industry. The 269 amino acid subtilisin from *Bacillus lentus* (Savinase) and the homologous subtilisins from *B. alcalophilus, B. amyloliquefaciens* and *B. subtilis* are the most studied proteins from this class (*Betzel et al., 1992*; *Remerowski et al., 1994*; *Martin et al., 1997*; *Graycar et al., 1999*; *Mulder et al., 1999*; *Radisky & Koshland, 2002*; *Tindbaek et al., 2004*; *Jones, 2011*). The subtilisins comprise a structural family with an $\alpha/\beta$ fold and a monomeric nearly spherical structure. The catalytic Asp/Ser/His triad is located in a groove on the surface of the protein. The substrate binds in two large pockets of the groove thus positioning the scissile bond in the active site. The substrate specificity is low but hydrophobic residues are favoured at positions P1 and P4 of the substrate (*Schechter & Berger, 1967*). In the structure of Savinase, two structural metal binding sites are found to be occupied by $Na^+$ and $Ca^{2+}$ (*Frankaer et al., 2014*).

Analysis of fast protein dynamics by NMR spectroscopy previously demonstrated that the structure overall is very rigid, but that the peptide segment 125–128, which is involved in substrate binding, is dynamic both on the time scale faster than nanoseconds as well as on the time scale of milliseconds (*Remerowski et al., 1996*). In the same study increased dynamics were also observed for residues 103 and 104 that cover the same part of the substrate binding site as the 125–128 segment and which is thought to be important for controlling substrate access to the binding site. These observations were supported by NMR relaxation and hydrogen exchange measurements on the homologous subtilisin PB92 (*Martin et al., 1997*; *Mulder et al., 1999*).

With the improved sensitivity and resolution of techniques used for assessing protein dynamics, several examples have appeared where conformational changes have been linked to specific steps in the molecular mechanism of the protein function. NMR relaxation methods have been used to characterize the conformational changes during catalysis in for example dihydrofolate reductase (DHFR) (*Boehr et al., 2010*), cyclophilin A (*Eisenmesser, 2002*), adenylate kinase (*Wolf-Watz, Kay & Kern, 2005*) and trigger factor (*Saio et al., 2018*). By following the temperature dependence of chemical shifts, NMR may also be used to probe the existence of alternative states (*Williamson, 2003*; *Teilum, Poulsen & Akke, 2006*; *Doyle et al., 2016*). Both the relaxation and temperature dependence of chemical shifts are sensitive to states populated as little as 1% (*Williamson, 2003*; *Kay, 2016*). The development of methods to analyse electron density near the noise level has provided great insight into alternative conformational states of proteins and the conformational changes occurring during enzymatic turnover (*Fraser et al., 2009*; *Fraser et al., 2011*). Recent advances in liquid chromatographic systems, the sensitivity of mass spectrometers, and MS data analysis tools have made HDX-MS a sensitive and powerful tool to study conformational dynamics of proteins (*Engen, 2009*; *Jensen & Rand, 2016*; *Masson et al., 2019*). These advances have revived the use of hydrogen exchange for assessing local conformational dynamics with only minute amounts of proteins in solution (*Fast, Vahidi & Konermann, 2017*; *Trabjerg, Nazari & Rand, 2018*) as well as in membranes (*Möller et al., 2019*; *Nielsen et al., 2019*). Despite the great insight in the mechanism of many enzymes, the existence of detectable

conformational dynamics of importance for the catalytic mechanism is not universal (*Jensen, Winther & Teilum, 2011*).

Here we have asked the question if the conformational dynamics previously observed in Savinase by NMR spectroscopy represents the existence of a specific conformational state that potentially could be linked to the function of the enzyme. To probe the conformational heterogeneity, we have followed the temperature dependent chemical shift changes, measured changes in the local stability of the enzyme when an inhibitor is bound and finally analysed the difference in electron densities from X-ray data collected at both cryogenic and ambient temperatures. Our analysis reveals that Savinase overall has very little conformational dynamics except for the substrate binding loops controlling the access to the active site that are more mobile than the rest of the molecule.

## MATERIALS & METHODS

### Protein production
Commercially available Savinase (Novozymes A/S) was dissolved in 0.1 M dimethylglutaric acid, pH 6.5, 0.2 M borate, 2 mM $CaCl_2$ and run through a Sephadex G25 SEC column equilibrated in the same buffer. The sample was purified by ion-exchange chromatography on CM-Sepharose with a gradient from 0 to 100 mM NaCl in the above buffer as previously described (*Betzel et al., 1988*). The peak fractions were pooled and for storage the buffer was changed to 100 mM $H_3BO_3$, 10 mM 2-(N-morpholino)ethane sulfonic acid (MES), 2 mM $CaCl_2$, and 200 mM NaCl at pH 6 by ultrafiltration. The expression media for the $^{15}N$ (and $^{13}C$) labelled samples was exchanged with M9 with $NH_4Cl$ as the sole nitrogen source (and U-$^{13}C_6$ glucose as the sole carbon source).

### NMR sample preparation
For NMR, 10 $\mu$M Savinase was incubated with 10 mM PMSF in 20 mM $Na_2HPO_4$-Citrate, pH 7. The mixture was incubated at 4 °C for approximately 4 h and subsequently filtered through a 0.2 $\mu$m filter. The PMSF-inhibited Savinase was separated from unreacted PMSF, concentrated to 0.8 mM and the buffer exchanged into 10 mM $Na_2HPO_4/NaH_2PO_4$, 2 mM $CaCl_2$, 1 mM 4,4-dimethyl-4-silapentane-1-sulfonic acid (DSS), 10% $D_2O$, pH 7.

### NMR experiments
All spectra were recorded on a Varian Inova 750 MHz spectrometer and the data were processed by NMRPipe (*Delaglio et al., 1995*) and analysed using CCPNMR (*Skinner et al., 2016*). The backbone resonances for PMSF-Savinase were assigned from $^1H$-$^{15}N$ HSQC, HNCA, HNCACB spectra recorded by reference to the published assignments of Savinase (*Remerowski et al., 1994*; *Remerowski et al., 1996*; *Martin et al., 1997*).

The temperature titration experiment was performed for PMSF-Savinase on a Varian Inova 750 MHz spectrometer. One dimensional $^1H$ spectra with presaturation and $^1H,^{15}N$ HSQC spectra were recorded in 2 K intervals from 289 K to 307 K. The chemical shifts of the backbone amide protons at each temperature were measured to a precision of 0.001 ppm relative to DSS. First-order and second-order polynomial models were fitted to the $^1H^N$ chemical shifts versus the temperature. The $F$-test ($p < 0.05$) was applied to determine significant deviations from linearity.

## HDX-MS sample preparation

Samples of PMSF inhibited and isotope unlabelled Savinase were prepared as described above. Samples of uninhibited and isotope unlabelled Savinase were prepared by exchange of buffer into 300 mM $Na_2HPO_4/NaH_2PO_4$, pH 7 to remove the borate inhibitor. To start the HDX reaction, 50 µM Savinase was mixed 1:20 with HDX labeling buffer (20 mM Bis-Tris, 10 mM $CaCl_2$ in 99.9% $D_2O$, pD = 7.0) at 25 °C. At time intervals of 0.25 min, 1 min, 10 min and 60 min, the exchange reaction was quenched by transferring 50 µl of reaction stock into 50 µl HDX quench buffer (300 mM $KH_2PO_4/H_3PO_4$, pH = 2.3) at 0 °C. The deuterated samples were immediately frozen and stored at −80 °C until the LC-MS analysis was performed.

Fully deuterated samples of Savinase were prepared by incubation in 8 M deuterated GndHCl (95% $D_2O$) overnight, and the reaction was quenched under the same conditions as above. For preparing non-deuterated references, 50 µM Savinase were mixed with the HDX buffer prepared in $H_2O$ in a volume ratio of 1:24. The solution was diluted with the same volume of the HDX quench buffer to obtain the final sample. The fully deuterated and non-deuterated samples were stored at −80 °C until analysis.

## Global HDX-MS experiments

Global HDX-MS analysis was performed on a combined HDX-UPLC system with a Synapt G2 mass spectrometer (Waters). The temperature for the UPLC system was maintained at 0 °C and equipped with a C4 column trap (ACQUITY UPLC BEH C4 1.7 µm VanGuard column, Waters) and an analytical C4 column (ACQUITY UPLC BEH C4 1.7 µm column, Waters). The protein sample was desalted initially on the trap column with a flow rate of 200 µl/min mobile phase A (0.23% formic acid in $H_2O$) for 3 min to remove buffer additives that may interfere with ionization. The protein was eluted with a 7 min gradient from 8% to 40% mobile phase B (0.23% formic acid in acetonitrile) for the ESI-MS at the rate of 40 µl/min. The generated positive ions were analysed by the mass spectrometer in MS-only mode, with the scan range from 50–2,000 m/z per 1 s. Global HDX experiments were performed in duplicate.

## Local HDX-MS experiments

Local HDX-MS analysis was performed on a combined HDX-UPLC system with a Synapt G2 mass spectrometer (Waters). The temperature for the HDX-UPLC unit was maintained at 0 °C and the system was equipped with a C18 column trap (ACQUITY UPLC BEH C18 1.7 µm VanGuard column) and an analytical C18 column (ACQUITY UPLC BEH C18 1.7 µm column). Deuterated and undeuterated proteins were digested on-line at 20 °C as they passed the manually packed pepsin column (Pepsin agarose, Pierce). The resulting peptic peptides were desalted on the trap column with the flow rate of 200 µl/min mobile phase A. Peptic peptides were eluted with a 12 min gradient from 8% to 40% mobile phase B to the mass spectrometer for positive mode ESI at the rate of 40 µl/min. For peptide mapping, the mass spectrometer was operated in MSe (DIA) mode with CID fragmentation. The scan range was set to 300–2,000 m/z for MS and 50–2,000 m/z for MS/MS, with the scan time of 0.5 s. For deuterated samples the mass spectrometer was

operated in MS mode with a scan range of 50–2,000 m/z and a scan time of 0.3 s. Local HDX experiments were performed in triplicate.

## Data analysis

For intact mass analysis (global HDX-MS), the acquired spectra were deconvoluted by MaxEnt in MassLynx 4.1 (Waters), and the resolution of deconvolution was set to 0.1 Da. For local HDX-MS analysis, peptides were identified by database searching using PLGS 2.5 (Waters). Analysis of peptide deuterium levels was performed using DynamX 3.0 (Waters). The final HDX processed data were exported directly from DynamX 3.0 to Pymol (Schrödinger) for illustration. The normalized deuterium uptake fraction is estimated by:

$$\text{Normalized HDX (\%)} = ([m_{\text{D}} - m_{\text{Non\_D}}]/[m_{\text{Full\_D}} - m_{\text{Non\_D}}]) \times 100\%.$$

Where $m_{\text{D}}$, $m_{\text{Non\_D}}$, and $m_{\text{Full\_D}}$ are centroid masses of deuterated peptides at each time point, undeuterated peptides, and fully deuterated peptides under the current experimental settings, respectively. To allow access to the HDX data of this study, the HDX Data Summary Table (Table S1) and the HDX Data Table (Table S2) are included in the Supplemental Information according to the community-based recommendations (*Masson et al., 2019*).

## Crystallization and diffraction experiments

The crystals for both room temperature (RT) and cryogenic datasets were picked from condition G2 of the PACT screen (*Newman et al., 2005*), 0.2 M NaBr, 0.1 M Bis Tris propane pH 7.5, 20% PEG3350. X-ray diffraction data were collected at the Diamond Light Source beamline I03. The ambient temperature (RT data) were collected at 297 K, and showed negligible radiation damage; the cryogenic data (cryo data) were collected at 100 K. Images were integrated using XDS (*Kabsch, 2010*) and scaled with Aimless (*Evans & Murshudov, 2013*).

## Initial structure calculation and refinement

Molecular replacement was performed using the published 1.4 Å resolution crystal structure of wild-type Savinase (PDB accession code: 4CFY) as the search model (*Frankaer et al., 2014*). Refinement was performed by PHENIX (*Liebschner et al., 2019*), to establish preliminary single-conformer structures for the RT and cryogenic crystals.

## Multi-conformer ensemble generation by qFit

For both the RT and the cryogenic data and starting from the single-conformers structures, qFit (*Keedy, Fraser & Van den Bedem, 2015*) was run to automatically construct multi-conformer models based on the observed electron density. A few manual adjustments were made to the models after the automatic qFit model generation. Five additional cycles of structural optimizations were performed using Phenix.refine with occupancy refinement and automatic solvent-picking. No hydrogen atoms were added. The models were refined with anisotropic B-factors except for water molecules.

## Ringer analysis for X-ray electron density maps

The optimized $2mF_{\text{o}}\text{-}DF_{\text{c}}$ density maps for single-conformer models were calculated using Phenix.maps with a grid spacing set to 0.2 of the resolution. The averaged side-chain

conformation of the single-conformer models for each residue was used as input for Ringer analysis (*Fraser et al., 2009*; *Lang et al., 2010*). The side-chain torsion angles were incremented in 10° steps and compared to the electron density above $1\sigma$ to build alternative side-chain conformations. The resultant multi-conformer models with alternative side-chain conformations were optimized using Phenix.refine, with the same refinement settings as for the refinement of the qFit models described above. The electron densities for residues with alternate conformations after the qFit and Ringer analyses were inspected in COOT and only those with convincing evidence for the alternate conformations were accepted for the final models.

## RESULTS

The broad specificity of Savinase makes it a substrate for itself. To prepare samples that are stable long enough for our NMR measurements it was necessary to inhibit the enzyme irreversibly. Using PMSF, we prepared homogenous samples that were stable for weeks. To complement the previous NMR relaxation and hydrogen exchange analysis we initially used NMR spectroscopy to follow temperature induced changes in chemical shifts. The temperature coefficients of the amide protons, $^1H^N$, are sensitive to the presence of hydrogen bonds. Thus $^1H^N$ that engaged in hydrogen bonds have d $\delta$/dT more negative than $-4.6$ ppb/°C (*Baxter & Williamson, 1997*; *Tomlinson & Williamson, 2012*). In line with this, the least negative d $\delta$/dT values are found in the loops and near the ends of $\alpha$-helices and $\beta$-strands. There is a tendency that the d $\delta$/dT values on the face of the protein around the active site are slightly larger than on the opposite side suggesting that the protein is most flexible around the active site (Fig. 1). In addition to be used as a measure of the strength of hydrogen bonds, the temperature dependence of the chemical shifts may also probe the existence of low populated alternative states. Divergence from linearity in plots of $\delta$ vs. T may thus result from a temperature dependent shift in population of such alternative states (*Williamson, 2003*; *Tomlinson & Williamson, 2012*). In Savinase, most residues show linear temperature dependence of the chemical shifts, but for 14 residues there is significant (*F*-test, $p < 0.05$) but subtle curvature in the $\delta$ vs. T plots (Fig. S1). These residues are, however scattered apparently randomly over the protein. The NMR temperature titration suggests that Savinase is most flexible around the active site (Fig. 1C), but no distinct alternative states were identified.

For several enzymes it has been reported that conformational dynamics may be attenuated by binding of inhibitors (*Boehr et al., 2010*; *Masterson et al., 2012*). We thus turned to HDX-MS to assess if there is any change in the local stability when Savinase is inhibited by PMSF. In addition to the low amount of protein needed, the advantage of HDX-MS is that the experiment time is short. Autoproteolysis is thus minimal in uninhibited Savinase and the HDX pattern can be compared between active and inhibited enzyme.

Firstly, the total global HDX was measured on samples of uninhibited and PMSF-inhibited Savinase. The HDX was very slow with 40% and 42% of the exchangeable hydrogens replaced by deuterium, respectively, after 60 min of incubation at pH 7 and
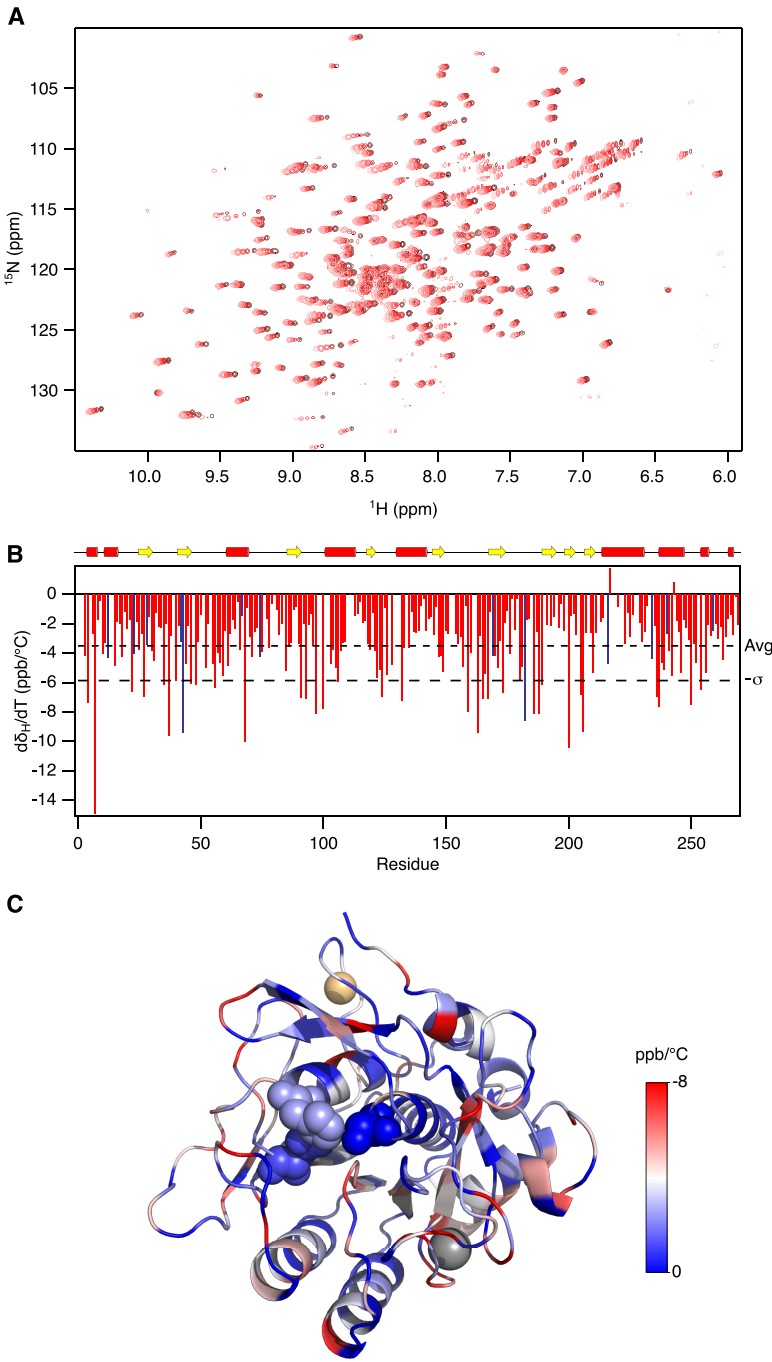

**Figure 1** **Temperature induced chemical shift changes in Savinase.** (A) $^{15}$N-HSQC spectra recorded at 18 °C, 22 °C, 26 °C, 30 °C and 34 °C in colours from light to dark red. (B) Temperature coefficients for the backbone amide proton chemical shifts plotted against sequence number. The dashed lines show the average temperature coefficient (Avg) and one standard deviation ($-\sigma$). (C) The structure of Savinase is colour coded according to the temperature coefficients shown in panel B. The colour scale is shown to the right of the structure. The metals as shown as spheres, $Ca^{2+}$ (grey) and $Na^+$ (orange). The catalytic triad in the active site is shown with spheres.

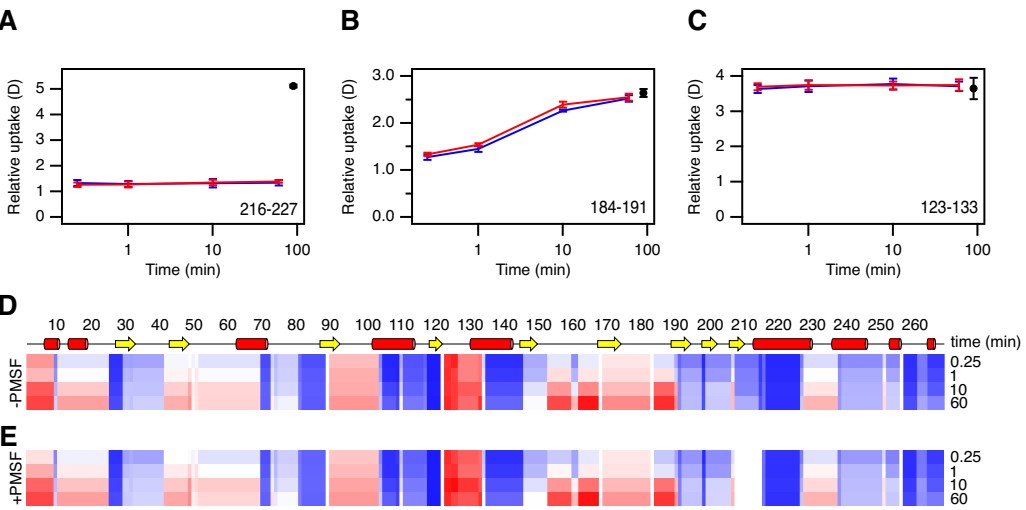

**Figure 2  HDX analysis of Savinase.** (A–C) Time dependent changes in deuterium content for three peptide fragments, as indicated in the lower right corner of each graph. Data for uninhibited and inhibited Savinase are shown in red and blue, respectively. Maximum labelled controls are shown in black. Error bars indicate standard deviations for time points measured in replicates ($n = 3$). (D–E) Heat map overview of the normalized local HDX of different regions. The two maps are for uninhibited Savinase (D) and inhibited Savinase (E). The normalized HDX for the four time points indicated on the right are displayed for each peptide segment experimentally resolved on a colour scale from no HDX (blue) over white to full HDX (red).

25 °C where hydrogen exchange is favoured (Fig. S2). Savinase thus has a stable core that rarely visits an open state from which HDX can occur. This is in agreement with the crystal structure and what would be expected from hydrogen exchange measured by NMR on the homologous PB92 subtilisin (*Fogh et al., 1995*).

To assign the HDX to different regions, Savinase was treated with pepsin before the MS analysis. This treatment yielded 110 peptides covering 100% of the sequence of the uninhibited enzyme, and 106 peptides covering 97% of the sequence of PMSF-inhibited enzyme. In this way, local HDX profiles for uninhibited Savinase and PMSF-Savinase were generated covering the full sequence, except residues 209-216 in PMSF-Savinase (Fig. 2). The resolution of the HDX data varies from single amides to stretches of 14 amides. In parts of the sequence, exchange occurs on the minute time scale. However, the segment from residue 123 to residue 133, where several labile hydrogens engage in hydrogen bonds in the structure, has almost fully exchanged within 15 s suggesting that this part of the protein is flexible. At the other extreme, there are segments that show almost no exchange within 60 min, particularly evident for the segments 119–122 and 217–227. As expected, most of the slow exchanging regions include residues involved in secondary structures in the interior of the protein.

Overall the local HDX profiles of uninhibited Savinase and PMSF-Savinase (Fig. 2 and Fig. S3) are very similar. We compared the data for the two situations by individual $t$-tests at each time point to determine if the deuterium uptake was significantly different. The results show only two minor differences were significant ($p < 0.01$) and that these were

assigned to two peptide segments 147–168 and 184–190/191. These reductions in HDX upon PMSF binding were however very minor (Fig. 2 and Fig. S3) and borderline with the normal limit of detection/confidence limit of the method (*Trabjerg, Nazari & Rand, 2018*; *Hageman & Weis, 2019*). The two segments 147-168 and 184-191 are located spatially close to each other and to the active site and thus could indicate a subtle local impact on the backbone dynamics in this region upon PMSF binding nearby.

It can therefore be concluded that modification of Savinase with PMSF has only a minor effect on the dynamics of the backbone that leads to exposure of amide groups. Hence, our NMR experiments performed on PMSF-inhibited Savinase should be indicative of the properties of the uninhibited native state of the enzyme. We note that the HDX-MS experiments were constrained to relatively short timescales due to issues with auto-proteolysis of the enzyme at longer timescales. HDX performed at longer timescales could potentially reveal differences between WT and PMSF-inhibited Savinase not detected here.

We next recorded independent sets of X-ray diffraction data recorded at room temperature and at cryogenic temperature. It has been demonstrated that cryo-conditions in some cases can favour conformations that are unfavourable at room temperature (*Halle, 2004*) and in other cases it has been demonstrated that minor conformations can be identified in high resolution electron density maps recorded at room temperature, which were not seen in the cryogenic data (*Fraser et al., 2009*; *Fraser et al., 2011*). Savinase was crystalized in space group P2$_1$2$_1$2$_1$ and diffraction data were collected to 1.1 Å and 0.95 Å resolution for the RT and cryogenic crystals, respectively. qFit and Ringer were used to construct multi-conformer models of Savinase for both the RT and cryogenic crystals (Table S3). The structures derived from the crystals at room and cryogenic temperatures are very similar, with an RMSD between the structures of 0.20 Å and 0.37 Å for the backbone and all heavy atoms, respectively. The average C$^\alpha$-C$^\alpha$ displacement of the aligned structures is 0.18 Å and the largest displacement is 0.80 Å. Thus, the two structures show no regions with major differences.

From the multi-conformer refinement, we find alternative conformations for 28 residues in the RT structure and for 25 residues in the cryo structure (Table S4). These alternative conformations are different sidechain rotamers except for the peptide segment 128-132 which in the cryo structure shows an alternative mainchain conformation (Fig. 3A). Most of the residues with alternative rotamers are serines (Figs. 3B–3C), 14/28 and 12/25 in the RT and cryo structures, respectively. Except for the peptide segment 128-132 in the cryo structure and the dipeptide 103-104 in both structures, the residues with alternative conformations are isolated in the structure and do not appear to engage in any cooperative network of conformational change.

## DISCUSSION

We have demonstrated that Savinase, except for alternative rotameric states of 25 sidechains is well described by a single conformation for the RT structure. In the cryo structure residues 128–132 have an alternative mainchain conformation and this segment also appears particularly flexible from the HDX-MS analysis. We note that this segment is
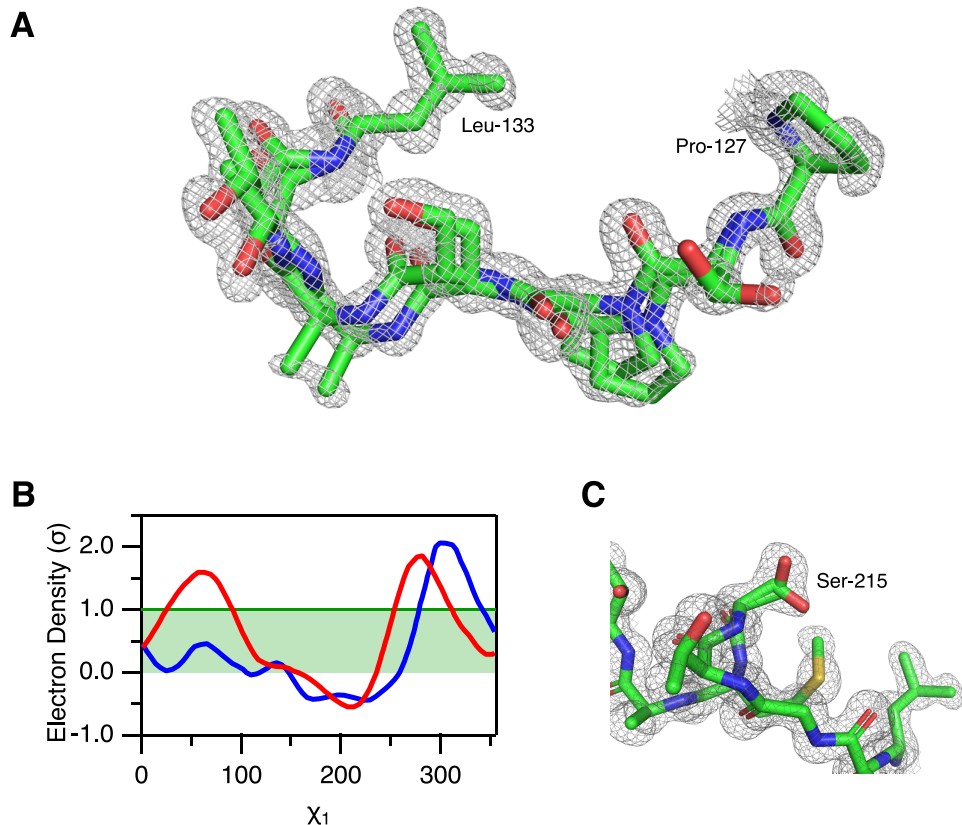

**Figure 3  Alternative conformations in the electron density of Savinase.** (A) Main chain heterogeneity in the cryo data of the peptide segment 128–132 with the $2mF_o\text{-}DF_c$ electron density map plotted at $1\sigma$. Residues 127 and 133 that only show a single conformation are also shown in the figure. (B) Ringer analysis of rotamers of Ser-215. The electron density is plotted as a function of the side-chain $\chi_1$ dihedral angle for the RT data (red) and the cryo data (blue). (C) Structure of Ser-215 and surrounding residues with the $2mF_o\text{-}DF_c$ electron density map of the RT data plotted at $1\sigma$. Two rotamers of the Ser-215 side chain with occupancies of 73% and 27% have been modelled to the electron density.

adjacent to the segment 125–128, which together with residues 103 and 104 that have alternative conformations in both the RT and cryo structures appeared more dynamics than the remainder of the protein from NMR relaxation analysis (*Remerowski et al., 1996*). The loops lining the substrate binding site have more negative chemical shift temperature coefficients, and even more clearly these parts of the protein have the largest crystallographic B-factors (Fig. 4). The B-factors, the temperature coefficients of the chemical shifts and the hydrogen exchange report on motions in the protein at different time-scales and will therefore have maxima in different regions of the protein. This is similar to the situation in other proteins where motions on different time-scales are observed in different regions of the proteins (*Pandya et al., 2018*; *Verteramo et al., 2019*).

The X-ray structures suggest that there was little to be gained through the qFit and Ringer analysis of Savinase due to its rigidity, perhaps with the exception of the easy building of the 128-132 conformation in the cryo-temperature structure. However, the

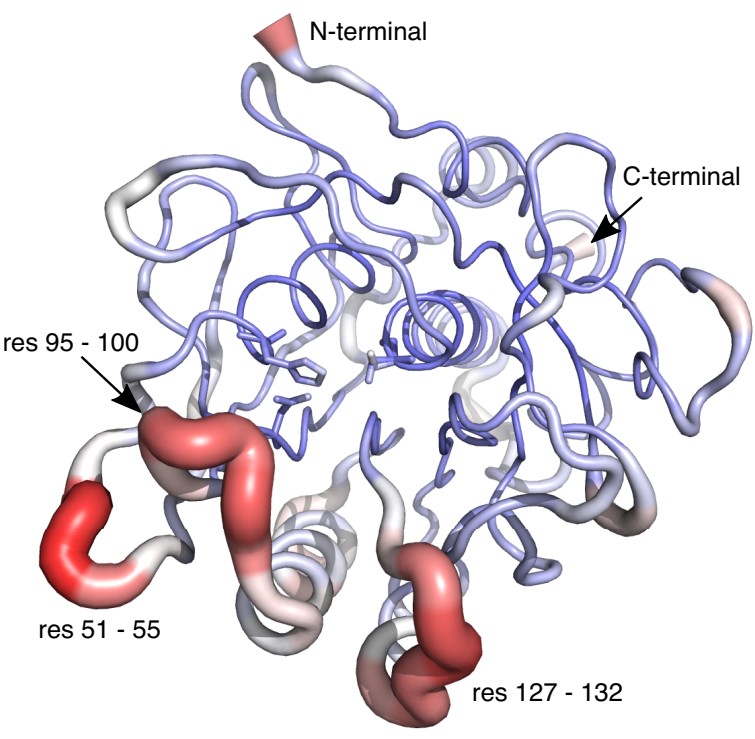

**Figure 4 Structure of Savinase from the crystallographic data collected at room temperature.** The width of the backbone trace is scaled with the B-factor. The colour scale is from blue (0 Å²) over white to red (35 Å²). The residues of the catalytic triad are shown as sticks.

latter is not seen in the isomorphous cryogenic 4CFY structure already in the PDB. The alternate conformations of side chains were generally all visible in a conventional refinement without qFit and Ringer analysis.

From the current study it was not possible to get additional insight into the structural dynamics that were suggested from previous NMR relaxation data and assumed to include conformational changes on the millisecond time scale around the substrate binding site. Such slow conformational changes have for many enzymes been linked to either substrate binding or the catalytic mechanism and we found it reasonable to hypothesise that such conformation changes would also be present in Savinase, in particular around the substrate binding site as suggested by NMR relaxation (*Remerowski et al., 1996*). However, the absence of alternative conformations is consistent with the very similar conformations of the homologue subtilisin BPN' with and without the inhibitor protein chymotrypsin inhibitor 2 (*Gallagher et al., 1996*; *Radisky & Koshland, 2002*). Even the loops covering the active site have almost identical conformations in these two structures.

Intrinsic microsecond-millisecond motions in e.g., adenylate cyclase are associated with large loop movements although conformational changes in e.g., protein tyrosine phosphatase 1B, RNase A and dihydrofolate reductase are of smaller amplitude (*Beach et al., 2005*; *Boehr et al., 2010*; *Whittier, Hengge & Loria, 2013*). Collectively these fluctuations allow for substrates and products to access or leave the active site. In these enzymes the

conformational changes necessary for binding and release of the substrate and product are rate-limiting for catalysis. In Savinase, the active site is more accessible, and no substantial structural changes are needed for substrate binding. This behaviour is similar to human glutaredoxin (*Jensen, Winther & Teilum, 2011*) and may explain why no conformational exchange is observed in the ground state of Savinase and suggest that substrate binding follows an induced fit like mechanism.

## CONCLUSIONS

Our aim was to assess if the conformational dynamics previously observed by NMR relaxation gave rise to a well-defined minor conformational state. The qFit and Ringer analysis of the crystallographic data did not identify significant minor conformations in this rather rigid enzyme structure. The other techniques applied also did not identify such alternative states, although there is more flexibility in the loops surrounding the active site than in the remainder of the protein. However, very low populated conformations that require more sensitive methods like NMR relaxation (*Kay, 2016*) and single molecule FRET (*Hatzakis et al., 2012*) can of course still be present.

## ACKNOWLEDGEMENTS

KDR and KT are members of Integrative Structural Biology at the University of Copenhagen (http://www.isbuc.ku.dk). Diamond Light Source is acknowledged for access to beamlines I03 and I04 (proposal number mx-9948) that contributed to the results presented here. The authors thank Sam Hart for assistance during data collection.

### Funding

This work was supported by the Danish Council for Independent Research (No. DFF-1335-00317 to KT and DFF-4184-00537A to KDR). The funders had no role in study design, data collection and analysis, decision to publish, or preparation of the manuscript.

### Grant Disclosures

The following grant information was disclosed by the authors:
Danish Council for Independent Research: DFF-1335-00317, DFF-4184-00537A.

### Competing Interests

Jens E. Nielsen is employed by Novozymes A/S.

### Author Contributions

- Shanshan Wu and Tam T.T.N. Nguyen performed the experiments, analyzed the data, prepared figures and/or tables, authored or reviewed drafts of the paper, and approved the final draft.
- Olga V. Moroz and Johan P. Turkenburg performed the experiments, authored or reviewed drafts of the paper, and approved the final draft.

- Jens E. Nielsen conceived and designed the experiments, authored or reviewed drafts of the paper, and approved the final draft.
- Keith S. Wilson, Kasper D. Rand and Kaare Teilum conceived and designed the experiments, analyzed the data, prepared figures and/or tables, authored or reviewed drafts of the paper, and approved the final draft.

## Data Availability

X-ray data are available together with the structures from RCSB Protein Data Bank (PDB): 6Y5T and 6Y5S. The HDX MS data are available in Table S2.

## Supplemental Information

Supplemental information for this article can be found online at http://dx.doi.org/10.7717/peerj.9408#supplemental-information.

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
