# Peer review of "Conformational heterogeneity of Savinase from NMR, HDX-MS and X-ray diffraction analysis"

_PeerJ, doi:10.7717/peerj.9408_

## Round 0.1 · original submission · Minor Revisions

Three reviews have been received, all overall positive, and contain some food for thought.

Reviewer 1 ·

Basic reporting

The article is clear and well written. Need to change line around 35 HDX does not exclusively report on H-bonding. I think there are too many citations, 56 I think for a short paper is too many in my opinion and needs to be focused more on those that are absolutely necessary. I think figure one needs more annotation as it stands it's not possible to see what the authors are referring to in the main text. In particular figure 1b needs a ribbon above it showing the secondary structure (they have provided similar in 2b) and they should highlight residues in the active site. Figure 1c also needs annotation with some arrows pointing to salient structure referred to in the main text. I also think there is a mistake in the gradient bar should go from -8 to 0.

Experimental design

I think the experiments are rigorous and well explained but I find it a shame that the authors have only used HDX-MS to underpin the NMR rather than use the HDX to look for rare states. In the conclusions the authors point to alternative methods that are more sensitive to rare states. However HDX-MS is sensitive to rare conformations but they haven't employed this technique to look for them. The longest labelling time is quite short and I appreciate they have concerns about auto-lysis in the enzyme. But with these short labelling times much of the protein doesn't exchange could there not be some slow EX1 exchange at longer times? Can the authors comment as to why they haven't incubated this protein for long enough to explore EX1 dynamics and potentially unearth a rare state?

Validity of the findings

The findings are not particularly interesting but I still think these types of data have value and have been conducted rigorously. I think the data that has been provided is sufficient to support the conclusions except the lack of long labelling times in the HDX data.

·

Basic reporting

The paper is well written and highly intelligible, with an introduction that clearly and interestingly focuses on the research question with exhaustive references to previous studies and to the techniques implemented. I would only suggest minor adjustments:
1. Specify that room temperature is 25°C for HDX-MS experiments (line 129) since for diffraction experiments it is set to a different value, namely 297K, i.e. 24°C (line 185).
2. When first using the abbreviation RT (line 182), you could specify that it stands for “room temperature”.

Thanking you for providing the raw data following the recommendations stated by Masson et al. (2019), I have to highlight that they show a couple of inconsistencies:
1. In Table S1, the number of replicates is set to 3 while the paper states that (local) HDX experiments were performed in tetraplicate (line 166). In second place, please specify in Table S1 that the units of HDX time course is minutes and correct the sequence coverage (one extra % is present).
2. In Table S1 the number of peptides is set to 106, while in Table S2 110 peptides are present and 112 peptides are declared in the manuscript for the uninhibited enzyme (line 252).
3. In Table S2 the last 32 peptides do not have associated values in the columns “Uptake – Control Sample (D)” and “Uptake error (SD) – Control Sample (D)”. Moreover, it could be nice to have a uniform value for non-performed measurements: in fact, sometimes an empty space is left, other times “NA” or “NaN” are used.

Figures are clear and well captioned, the only typo to be corrected is in the caption of Figure 3-C (it is referred as B instead of C).

Experimental design

No comment.

Validity of the findings

I would strongly suggest inserting the plots showing the results of the analysis described in lines 231-233 of the manuscript.

I disagree with the statement in lines 256-257: “In the most parts of the sequence, exchange occurs on the minute time scale”. I would remove the "most" since looking at Figure S2 the “no exchange” and the “minute time scale” situations are comparable in terms of probability to occur (maybe the "no exchange" situation is even more probable). As no exchange occurs for segments 119-122 and 217-227 (line 264), I would say that no exchange occurs even for other segments like 1-30, 1-31, 1-41, 2-31, 3-31. Anyway, this does not influence the overall conclusion.

·

Basic reporting

In this study, an important enzyme for biological washing applications – savinase – is analysed to answer the hypothesis that it has a specific conformation which confers activity. Multiple biophysical approaches are brought together to yield sub-molecular resolution of structural- and thermo-dynamics: X-ray crystallographic structural models, 1-D and 2-D NMR and hydrogen/deuterium-exchange mass spectrometry. The study provides two alternative active and inactive conformations of the savinase structure and these are annotated with detail of the extent of conformational dynamics, resulting from covalent inactivation by PMSF irreversible inhibition. The conclusion of the study is that there are no significant differences in local or global structure to support the hypothesis that there is an alternative active conformation of savinase.

Communication is excellent.
The study is well connected to the immediate and wider literature. Citations were full and complete, as fast as I could determine.
Enough raw, or processed data has been shared.

Experimental design

The authors perform Student t-tests per peptide per timepoint, which is a very robust statistical treatment of the data. This is crucial to the overall objective here of sensitively comparing two states of a highly stable enzyme and, in particular, to support the finding that there are no significant differences. The study has been performed experimentally and data statistics performed to a high level.

Detail of methods is absolutely fine, as far as I can tell. Certainly enough experimental detail is provided that the experiments could be replicated successfully.

Validity of the findings

Conclusions are supported by the data, with only minor comments (see below note).

Additional comments

Major comments:

No major criticisms found.

Minor Comments:

1. The NMR and MS methods employed here are ensemble methods. Please can the authors comment on the extent that their measurements conclusively refute the existence of a sub-population of the alternative conformation that was indicated by NMR relaxation data. What fractional population would be required in order to be detected above noise in this dataset?

2. I recommend adding a volcano plot of the HDX-MS data, as this would make for a simple communication of the hybrid significant differences between the savinase states – or lack thereof. This was shown to great effect in Hageman & Weis which the authors have cited here.

3. HDX data for some peptides is absent or incorrectly isotopically assigned for the PMSF state in S2:
• 208-226
• 209-226
• 209-227

4. Page 14 line 289: Quantify “major differences” between crystal models. Is there a threshold of Å displacement that qualifies as a major difference?

5. Mentioned in a few places (e.g. Page 13 line 233): What was the non-linear model used in the F-test? Can the authors please provide a supplementary figure that locates the pseudo-random distribution of these deviating residues (e.g. highlighted on the structure)? Perhaps it is meant to be clear from Fig. 1, but I didn’t find this the case.

6. Figure 3 Legend: “side-chain ꭓ1 dihedral angle” formatting error.

---

## Round 0.2 · accepted · Accept

Thank you for revising your paper. Please check the outstanding comment from Reviewer 2, asking, if possible to add data to supplementary material.

Reviewer 1 ·

Basic reporting

Everything I asked for has been updated I have no further issues

Experimental design

Everything I asked for has been updated I have no further issues

Validity of the findings

Everything I asked for has been updated I have no further issues

·

Basic reporting

Communication in the paper is optimal: the article is highly intelligible and well referenced.
About HDX-MS raw data (Supplementary Table S2): the fully deuterated reference sample is missing for the last 32 peptides. I strongly recommend to insert these data, otherwise such peptides could not be used by an external user. In fact, normalization expressed in line 179 would not be possible.

Experimental design

The research question is well defined and opportunely addressed in lines 89-97, providing further proof of the clarity of the paper.
The different methods implemented to probe Savinase testifies a rigorous investigation of the protein's dynamical behaviour.

Validity of the findings

The conclusions are well stated and the findings are supported by a sufficient amount of data.

Additional comments

The paper is well written, the research question is clear and well addressed.
The only critisism I have is about HDX-MS data (Supplementary Table S2) since uptake values for the reference samples are missing for the last 32 peptides. Once these values have been added, the paper deserves to be published.

·

Basic reporting

no comment

Experimental design

no comment

Validity of the findings

no comment

Additional comments

I recommend that this manuscript is acceptable for publication following the amendments.